# *Bacillus cereus* biovar *anthracis* causes inhalational anthrax-like disease in rabbits that is treatable with medical countermeasures

**Allison M. Ferris, David G. Dawson, Andrea B. Eyler, John J. Yeager, Jordan K. Bohannon, Jeremy A. Boydston, Melissa L. Krause, Charles L. Balzli, Victoria Wahl**[ORCID]**, Tammy D. Jenkins, Sherry L. Rippeon, James E. Miller, Susan E. Miller, David W. Clarke, Emmanuel Manan, Ashley F. Harman, Kim R. Rhodes, Tina M. Sweeney, Heather D. Cronin, Ron L. Bowman, Michael P. Winpigler, Heather A. Zimmerman, Alec S. Hail, Angelo Scorpio**[ORCID]*****

National Biodefense Analysis and Countermeasures Center, Frederick, Maryland, United States of America

* angelo.scorpio@st.dhs.gov

## Abstract

*Bacillus anthracis* is a zoonotic organism that causes the disease anthrax due to the activity of virulence factors harbored on plasmids pXO1 and pXO2. Inhalation of *B. anthracis* spores results in pneumonic disease that progresses quickly, and often results in lethality in the absence of medical countermeasure (MCM) intervention. Recently, reports have identified *Bacillus cereus* isolates that possess pXO1 and pXO2-like plasmids and cause an anthrax-like disease. These isolates have been named *B. cereus* biovar *anthracis*, or Bcbva. To evaluate disease course of Bcbva, the inhalational median lethal dose (INHLD$_{50}$) was determined for two isolates, Bcbva Cameroon (CA) and Bcbva Cote d'Ivoire (CI), using the New Zealand white (NZW) rabbit inhalation anthrax model and compared to established *B. anthracis* inhalation data. Furthermore, disease progression and anthrax MCM efficacies were evaluated by quantifying temperature responses, bacteremia, and virulence factor production in both survivor and non-survivor animals. This study determined that the rabbit INHLD$_{50}$ values for Bcbva CA and CI were similar to that published for *B. anthracis* Ames. The mean time to significant increase in body temperature (SIBT) and death were dose dependent for both Bcbva isolates, and all animals that succumbed to aerosol exposure displayed SIBT prior to death. Serum hyaluronic acid concentration increased prior to mortality in animals challenged with Bcbva and differences were observed in serum protective antigen concentration in animals challenged with Bcbva compared to *B. anthracis*. Pre-exposure vaccination with Anthrax Vaccine Adsorbed (AVA) and post-exposure prophylaxis of levofloxacin with or without AVA vaccination were effective against a challenge of ~200 INHLD$_{50}$ of Bcbva CA or CI. Collectively, these data suggest that anthrax-like disease caused by Bcbva is similar to that caused by *B. anthracis* Ames 2084, and that currently available countermeasures are effective against inhalation exposure to Bcbva.

**Data availability statement:** All relevant data are within the manuscript and its Supporting Information files.

**Funding:** VW received funding award All authors were under a federal contract. Funding was under Contract No. HSHQDC-15-C-00064 with the funder U.S. Department of Homeland Security Science and Technology Directorate. https://www.dhs.gov/science-and-technology The funders had no role in study design, data collection and analysis, or preparation of the manuscript. The funders were involved in the decision to publish. Authors salaries were covered under the funding contract, through Battelle National Biodefense Institute (BNBI)

**Competing interests:** The authors have declared that no competing interests exist.

## Author summary

*Bacillus cereus* biovar *anthracis* (Bcbva) is a spore-forming bacterium that causes anthrax-like disease in mammals. Utilizing the New Zealand white (NZW) rabbit inhalation model, this study demonstrates that aerosol exposure to Bcbva Cameroon (CA) or Cote d'Ivoire (CI) spores results in anthrax-like disease and mortality similar to that observed for *Bacillus anthracis*. The inhalational median lethal doses of Bcbva CA and CI, times to signs of disease, expression of virulence factors, and terminal bacterial burden were similar to that reported for *B. anthracis*. Biological samples taken during the course of infection and plated on agar media indicated that Bcbva CA and CI had variations in colony morphology depending on the sample time post-exposure. The time to a significant increase in body temperature (SIBT) and death was dependent upon inhaled dose, with SIBT predictive of death in all animals challenged with either strain. Anthrax medical countermeasure (MCM) efficacy was also evaluated in the aerosol exposure model. Pre-exposure vaccination with Anthrax Vaccine Adsorbed and post-exposure prophylaxis with levofloxacin (7-day treatment) with or without vaccination were effective treatments for Bcbva anthrax-like disease. This study will enhance our understanding of Bcbva, the anthrax-like disease that it causes, and inform treatment options.

## Introduction

*Bacillus anthracis*, the causative agent of anthrax, is a zoonotic organism that primarily causes disease in livestock, but has the potential to cause disease in humans that handle contaminated skins and furs or ingest meat from infected animals [1–4]. When observed in a global health context, *B. anthracis* annually has limited impact on human health [5]. However, for historical reasons, including the Amerithrax incident in the United States in 2001, *B. anthracis* is considered a biological threat agent. As a result, there have been significant investments in efforts to understand its biology, develop vaccines and treatments that prevent or eliminate infection, develop methods for environmental and clinical detection, and develop public health response plans [4,6,7].

Evidence of genetic and phenotypic diversity within the *Bacillus* genus [5,8–12] raises questions as to whether emerging species pose the same risks to human health as *B. anthracis*. There have been several reports on the isolation and characterization of species related to *B. anthracis* that cause an anthrax-like disease. Specifically, *Bacillus cereus* isolates have been discovered that possess pXOl and pXO2-like virulence plasmids, which in *B. anthracis* are required for toxin and capsule production, respectively [10,13–15]. These pXOl and pXO2-like plasmid-containing variants of *B. cereus* were isolated from a human (G9241) and great apes (*Bacillus cereus* biovar *anthracis*; Bcbva) following evidence of an anthrax-like disease. The ability of some *B. cereus* isolates to cause an anthrax-like disease (for Bcbva, henceforth referred to as anthrax) in animals has resulted in the formation of a biovar *anthracis* sub-group within this species, which is represented by five isolates from the Ivory Coast, Cameroon, Central African Republic and Democratic Republic of the Congo [15–19]. A retrospective human serum sampling study of 1,386 volunteers who live around Ta'i National Park in Cote d'Ivoire determined that 10.46% of the sampled individuals had antibodies against a Bcbva antigen despite no known cases of Bcbva-associated anthrax [20]. One review of over 50 natural Bcbva infections of animals in the Ta'i National Park concluded that Bcbva infection resulted in a form of anthrax with higher fatality rate than that caused by *B. anthracis*, and Bcbva infections resulted in a lower antibody response when compared to natural *B. anthracis*

infections [21]. Considering these public health data and because of the potential to cause lethal anthrax, these isolates have recently been categorized as select agents by the Centers for Disease Control and Prevention (CDC) [22].

A recent study determined the subcutaneous and intranasal lethal doses of Bcbva in mice and guinea pigs [17]. Like previous studies with *B. anthracis*, immunization of mice with recombinant protective antigen (rPA) and formaldehyde-inactivated *B. anthracis* spores conferred protection against a subcutaneous challenge with Bcbva. Vaccination with rPA alone was not protective against Bcbva, also consistent with previous *B. anthracis* studies [17,23]. To extend these findings, a rabbit inhalational anthrax model was used to assess lethality and disease progression following challenge with Bcbva Cameroon (CA) and Cote d'Ivoire (CI). We evaluated production of the hyaluronic acid capsule, poly-D-glutamic acid capsule, and the tripartite toxin during infection and the results were compared with historical data for the *B. anthracis* ancestor strain, Ames A2084. Additionally, MCM efficacy against Bcbva aerosol exposure was evaluated using pre-exposure vaccination with Anthrax Vaccine Adsorbed (pre-AVA) or post-exposure prophylaxis (PEP) with levofloxacin with or without AVA.

## Methods

### Ethics statement

All research was conducted in compliance with the Animal Welfare Act and other federal statutes and regulations relating to animals and experiments involving animals and adheres to principles stated in the Guide for the Care and Use of Laboratory Animals and approved by both the National Biodefense Analysis and Countermeasures Center Institutional Animal Care and Use Committee and, when applicable, the Department of Homeland Security Compliance and Assurance Program Office. The facility where this research was conducted is fully accredited by the Association for Assessment and Accreditation of Laboratory Animal Care, International (AAALAC, International) and all work was performed in compliance with a Public Health Service (PHS) Humane Care and Use of Laboratory Animals (Policy) Assurance.

### Bacterial strains and growth conditions

*B. cereus* biovar *anthracis* strains CA and CI were grown in a Biosafety Level 3 (BSL-3) laboratory in brain heart infusion broth (BHI; Remel) or on tryptic soy agar containing 5% Sheep's blood (SBA: Remel, Lenexa KS, USA). To induce expression of toxin and capsule, strains were grown in BHI containing 0.8% sodium bicarbonate and 5% carbon dioxide ($CO_2$), 37°C, shaking at 200 rpm for 16-18h. Images of the colony morphologies were documented with a Pentax WG-3 16.0-megapixel digital camera (Tokyo, Japan). To prepare spores for challenge studies, frozen stocks were diluted 1:1,000 in Leighton-Doi [24] in a 2 L baffled Erlenmeyer flask (200 mL final volume; Fisherbrand, ThermoFisher, Waltham, MA, USA) and incubated at 35°C with 225 rpm agitation utilizing a rotary shaking incubator for 72h, or until approximately 95% sporulation was reached as determined by phase contrast microscopy. The spores were harvested by centrifugation in 500 mL centrifuge bottles (Avanti JE centrifuge, JLA 10.5 rotor, 5,000 RPM, 20 min, 4°C) and washed twice with cold water for injection (WFI; Gibco ThermoFisher). A 58% solution of Gastrografin (Bracco Diagnostics, Milan, Italy) in WFI was prepared, and the spore suspension was gently layered onto the gradient followed by centrifugation in an Avanti JS 5.3 rotor (5,300 RPM, 30 min, 4°C). The spore pellet was washed three times with WFI, resuspended in WFI, and the spore preparation stored at 4°C. Prior to use, spores were heat-shocked at 65°C for 40 min. All spore preparations, prior to and post-heat shock, were enumerated by serial dilution in phosphate buffered saline (PBS) and plating on SBA.

## Polymerase chain reaction (PCR) characterization

PCR amplification of three gene targets for *B. anthracis* (*saspB*, chromosome; *pagA*, pXOl; and *capA*, pXO2) and a Bcbva target (genomic island IV [16]) were used to confirm the identity of the isolates and presence of virulence plasmids. Colonies cultured from terminal blood samples on SBA were resuspended in 0.5 mL of sterile PBS. A sample of each bacterial suspension (5 µL) was added to the well of a MicroAmp Fast Optical 96-well reaction plate (Applied Biosystems, Life Technologies; Waltham MA, USA) in duplicate. The reaction mixture containing Taqman Universal PCR master mix (15 µL; Applied Biosystems, Life Technologies), primers, and probes (Integrated DNA Technologies, Coralville, IA, USA) was added to each well containing sample, for a final volume of 20 µL, and the contents of each well were gently mixed by pipetting. The amplification reaction was performed in an Applied Biosystems 7500 Fast Real-Time PCR System (Life Technologies). Amplification of the target gene was analyzed using the 7500 Software v.2.0.5. The presence or absence of target gene amplification used a cutoff value of 35 cycles.

## Animal inhalational challenge system

The nose-only inhalation test system, housed in a class III biological safety cabinet (BSC III), was constructed as diagrammed in Fig 1, using an oronasal mask (Small Canine Mask, part# 32393B4, SurgiVet, St. Paul, MN, USA) and an aerosol delivery plenum assembled using stainless steel sanitary fittings (McMaster-Carr, Robbinsville, NJ, USA). A 3-jet Collison nebulizer was used to generate aerosols, and air samples were collected at one liter per minute on gelatin filters (SKC Gulf Coast, Eighty Four, PA, USA) to determine inhaled dose as previously described [25]. A continuous flow of HEPA-filtered house air was supplied to the nebulizer at 7.5 liters per minute to allow animals to breath normally. Aerosol particle size as mass median aerodynamic diameter (MMAD) was determined with an Aerodynamic Particle Sizer 3321 (TSI Inc., Shoreview, MN, USA) [25].

## Animals and temperature monitoring

New Zealand White rabbits (NZW, 3–4 kg, Charles River Laboratories, Frederick, MD, USA) were used to model respiratory infection and pneumonic anthrax as previously described [25]. All animals were surgically implanted with M00 digital transponders (Data Sciences International, St. Paul, MN, USA) in an intramuscular pocket within the peritoneal cavity as previously described [25]. After surgical recovery and prior to challenge, baseline temperatures of implanted rabbits were collected at one min intervals for two to three days. Following the baseline collection period, rabbits were removed from isolator cages [25], challenged by inhalation and immediately returned to the cages for continued temperature readings. Temperature readings continued until the rabbits either succumbed to infection or were euthanized. All animals were disposed of per standard ABSL-3 practices at the conclusion of the study. Data was analyzed by binning the temperature reads in 15 min intervals and plotting the corresponding mean temperatures against time.

When analyzing temperature responses, a significant increase in body temperature (SIBT) was defined as the mean baseline temperature plus three times the standard deviation of the mean baseline. Since antibiotic administration delays symptom onset, time to SIBT for antibiotic-treated animals was determined from the cessation of treatment. Time of death was defined as the time of euthanasia or a sudden non-reversed drop in body temperature. Peak SIBT was defined as the highest recorded temperature that was at least three standard deviations above the mean baseline. Mean SIBT in a group was defined as the mean of all recorded temperatures that were three standard deviations above the mean baseline. The number of

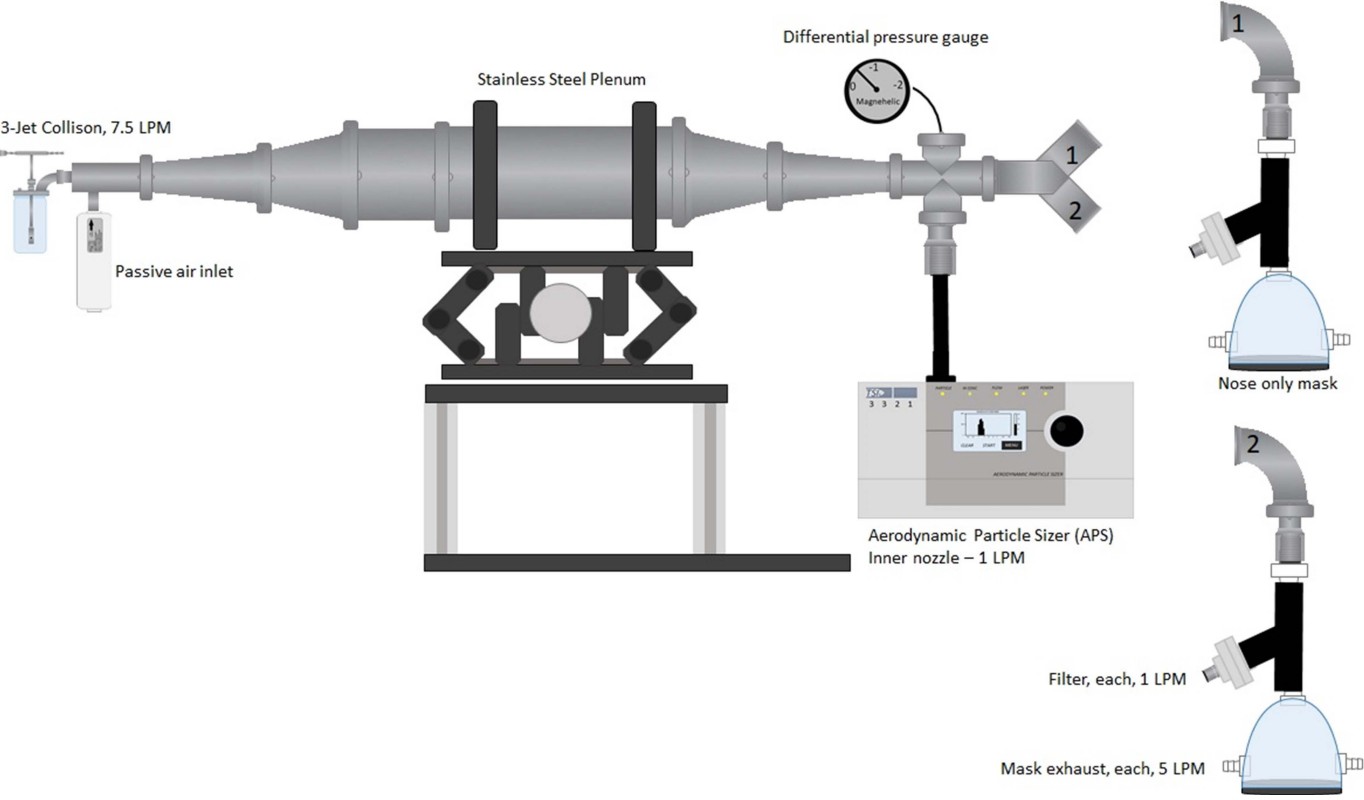

**Fig 1. Nose-only rabbit aerosol exposure test system.** The diagrammed laboratory-scale system was housed in a class III biological safety cabinet (BSC III) for safe operations. Animal exposure to aerosolized spores was performed as previously described [25]. A 3-jet Collison nebulizer was loaded with Bcbva spores diluted to an approximate dosage target. The test system was operated under passive negative pressure conditions in which air was continuously drawn through the system by the APS and vacuum from each mask (1, 2) during animal exposure. The aerosolized particles generated from the Collison were counted and sized by the APS and aerosolized spores quantified from filters after sampling the exposure air stream at one liter per minute. Image created in Microsoft PowerPoint using drawing tools in PowerPoint.

SIBTs was determined by counting the number of instances where temperature readings were higher than three standard deviations above the mean baseline for at least 30 min and subsequently returned to below three standard deviations above the mean baseline. SIBT duration was defined as the cumulative time the temperature readings were above three standard deviations above the mean baseline.

## Inhalation virulence in rabbits

Dose groups (Bcbva CA = 5, CI = 4) consisting of eight (four male and four female) NZW rabbits each were used to determine the median inhalation infectious dose (INHID$_{50}$ by presence of SIBT) and inhalation lethal dose (INHLD$_{50}$) and assess disease progression of each Bcbva strain. Prior to each test, the concentration of spores was adjusted to achieve the target inhaled dose and 10 mL transferred to the 3-Jet Collison nebulizer. NZW rabbits were anesthetized with acepromazine (0.25 - 1.0 mg/kg, 0.1 mL, Covetrus, Portland, ME, USA) for the baseline pre-bleed, placed in a cat sack physical restrainer (Covetrus), and transferred to the BSC III containing the inhalation exposure system (Fig 1). Rabbits were exposed to aerosols containing Bcbva spores for 10 min prior to removal from the BSC III. During the exposure, animals were monitored by veterinary technical staff for normal breathing. The inhaled dose was calculated as the product of the average aerosol concentration, the respiratory minute volume

(RMV) estimated from the body weight of the test subject, and the exposure duration; *Inhaled dose = $C_\alpha \times RMV \times Exposure\ Duration$* [25]. The relationship between respiratory minute volume and body weight described by Bide, et al. [26, 27] was used to estimate minute volume for each animal. Following exposure, animals were monitored for signs of infection a minimum of twice a day for 28 days. Animals showing signs of infection were humanely euthanized with a ketamine-xylazine cocktail (35 and 5 mg/kg, respectively; Covetrus) followed by an intra-cardiac injection of 3 mL Euthasol (Virbac, Westlake, TX, US) when meeting pre-determined criteria described in a clinical scoring sheet.

## Biological samples from exposed NZWs

Blood and serum samples were collected (immediately prior to challenge on day 0) and on days 1, 2, 3, 4, 5, 7, 10, 14, 21, and 28 post-challenge using a 23-gauge butterfly catheter inserted into either the central or marginal ear vein. Terminal sampling of animals that met euthanasia end points or were found dead was performed via cardiac puncture. Blood was placed in EDTA tubes, and immediately used to determine concentration by bacterial colony forming unit (CFU) enumeration. For all animals that succumbed to infection and for a subset that survived to the end of the study, terminal blood, lung, liver, spleen, and broncho-alveolar lavage (BAL; 10 mL) samples were also collected. Tissues were homogenized using an IKA Disperser (DT-20-M-Gamma Tube with Rotor-Stator, IKA, Wilmington, NC, USA) for 1 min in 10 mL of PBS. Bacterial titer in the homogenate was determined by serial dilution in PBS and plated on SBA as described above. Serum was isolated by collecting blood in BD Micro-tainer tubes (Becton Dickinson, Franklin Lakes, NJ, USA) followed by centrifugation (12,000 RPM, 1.5 min, RT), and samples were stored at -20°C for subsequent virulence factor expression analysis.

## Virulence factor expression and antibody response

Rabbit serum samples were evaluated for the presence of protective antigen (PA), soluble hyaluronic acid (hyaluronan, HA), and soluble poly-D-glutamic acid (PDGA). PA was quantified using a direct protective antigen ELISA kit per manufacturer's instructions (Alpha Diagnostics International, San Antonio, TX, USA). Soluble HA was quantified using a hyaluronan/hyaluronic acid ELISA kit per manufacturer's instructions (R&D Systems, Minneapolis, MN, USA). Soluble PDGA was quantified via sandwich ELISA utilizing a mouse monoclonal antibody raised against PDGA for both antigen capture and detection [28]. The detection antibody was labeled with alkaline phosphatase (AP) using the Surelink kit (SeraCare, Milford, MA, USA), and AP conversion of BluePhos was utilized as the substrate. All absorbance measurements were performed using a Spectra Max M5 (Molecular Devices, San Jose, CA, USA).

## MCM efficacy assessment

Eight groups of 20 animals (10 males and 10 females) were used to assess MCM efficacy against a target challenge dose of 200 x $INHLD_{50}$ of either Bcbva CA or CI. AVA was purchased from Emergent Biodefense Operations Lansing. An oral solution of levofloxacin (25 mg/mL) was purchased from a licensed pharmacy (Hi-Tech Pharmacal, Amityville, NY, USA). Control animals (n=20) did not receive any treatment or placebo. For PEP antibiotic-only treated animals, oral levofloxacin (50 mg/kg) was administered once daily, starting 6 – 8 h post challenge, for seven days. For PEP antibiotic and AVA-treated animals, levofloxacin was administered as described for the antibiotic-only treated animals, and 0.5 mL of undiluted vaccine was administered 6 – 8 h post challenge with a booster seven days later, both via intramuscular (IM) administration. Pre-exposure vaccinated animals were administered 0.5 mL of undiluted AVA

vaccine via IM administration followed by a 0.5 mL of undiluted AVA vaccine boost 28 days later. Animals were then challenged 70 days after initial vaccination.

In addition to PA and HA, rabbit serum samples were evaluated for the presence of anti-PA IgG using a rabbit anti-anthrax protective antigen 83 (PA83) IgG ELISA kit (Alpha Diagnostics International) per manufacturer's instructions. Serum samples from $INHLD_{50}$ experiments were used to measure seroconversion of untreated survivors and non-survivors. An increase in anti-PA83 IgG greater than three standard deviations of the background mean indicated seroconversion for PA83.

## Statistical analysis

Pairwise statistical comparisons were performed with the unpaired, two-tailed Student's T-test in either Microsoft Excel 2013 or GraphPad Prism 8.4 or higher. Analysis of multiple variables with Tukey's (Dunnett's for MCM studies) multiple comparisons test was determined by ANOVA analysis (GraphPad Prism 8.4 or higher). Mean times to death and SIBT were calculated using Kaplan-Meier survival analysis. Statistical analysis of dose effects on time to an event and MCM administration effects on time to an event was performed using the Mantel-Cox log rank test in GraphPad Prism 8.4 or higher. Since it was statistically determined that dose influenced time to SIBT and time to death (Mantel-Cox log rank test), a response screening model using Fisher's Exact Test was performed on animals that succumbed to infection to determine whether there was a sex bias with respect to these same parameters, and whether MCM administration influenced outcome (survival or non-survival) relative to the control group. Probit analysis was used to estimate the $INHID_{50}$ and $INHLD_{50}$ with 85% confidence intervals using JMP version 16. Dose was treated as continuous variable, and sex was treated as a categorical variable using JMP version 11 when determining factors impacting lethal dose. Statistical significance for the PEP levofloxacin-only group was evaluated at 6 days after cessation of treatment and on day 28 post-challenge, while dual vaccine and antibiotic post-exposure and pre-exposure vaccination group results were evaluated for statistical significance at day 28 post-challenge.

# Results

## Bcbva spore aerosol characteristics

A previously established inhalation model in which rabbits were exposed to *B. anthracis* Ames 2084 spores aerosolized with a 3-jet Collison nebulizer was used for aerosol exposure to Bcbva (Fig 1) [25]. The use of gradient purified Bcbva spores generated a repeatable, consistent, and well-defined particle size distribution with a MMAD range of 1.30 and 1.38 μm for Bcbva CA and CI, respectively, and a geometric standard deviation (GSD) of 1.43 and 1.35, respectively (Fig 2A).

## Bcbva CA and CI inhaled $ID_{50}$ and $LD_{50}$ estimation

Body temperature of rabbits exposed to aerosolized Bcbva spores was monitored and the times from exposure to presentation of SIBT and to death were recorded. SIBT was used as a marker for infection to calculate the $INHID_{50}$ and death was used to calculate the $INHLD_{50}$ by probit analysis. Bcbva CA (Fig 2B) and CI (Fig 2C) $INHLD_{50}$ with 85% confidence intervals were calculated as $2.16 \times 10^5$ [$1.22 \times 10^5$, $4.83 \times 10^5$] CFU and $3.48 \times 10^5$ [$2.25 \times 10^5$, $5.78 \times 10^5$] CFU, respectively. These values approximate the $INHLD_{50}$ of $1.05 \times 10^5$ CFU previously reported for *B. anthracis* Ames in the NZW rabbit inhalation model [29]. SIBT was used as an indicator of infection to determine the median infectious doses, and since all the animals that succumbed to Bcbva challenge had previously developed a SIBT and all animals that developed a SIBT succumbed to aerosol exposure, the median infectious and lethal doses are the same value for both Bcbva CA and CI.

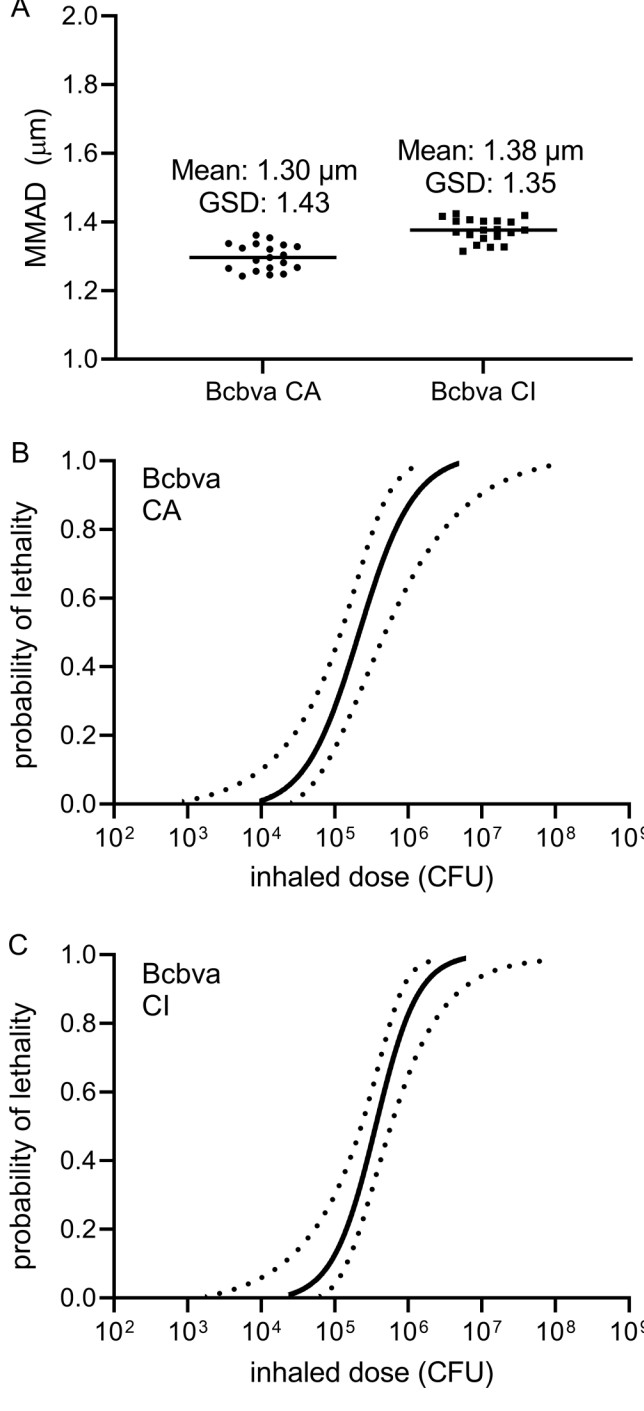

**Fig 2. Dose-response in rabbits exposed by inhalation to Bcbva CA and CI spores.** Particle size distributions of purified spores during aerosol exposures were measured by the APS. The mass median aerodynamic diameter (MMAD) for each set of animal exposures is indicated by individual dots (A). The horizontal line represents the replicate mean and corresponding geometric standard deviation. Probit dose-response (solid line) with 85% confidence intervals (dashed lines) are shown for Bcbva CA (**B**) and CI (C).

## Characterization of disease progression

Aerosol exposure to Bcbva CA or CI spores resulted in three characteristic temperature response patterns (Fig 3). Animals that survived aerosol exposure displayed no significant changes in body temperature over the course of observation (Fig 3). However, animals that succumbed to infection presented either a short (< 12 h) SIBT that was followed by death/ humane euthanasia, or a prolonged SIBT (> 24 h) that undulated in magnitude prior to death/ humane euthanasia (Fig 3). The protracted SIBT response was more commonly associated with lower inhaled doses. Most animals succumbed two to five days after challenge (Figs 4 and S1 Fig). However, there were several animals that succumbed outside of this range, with one animal succumbing 18 d after Bcbva CA challenge. Interestingly, none of the animals presented obvious signs of disease during observation except for the highest dose groups for each isolate. In the highest dose group, animals displayed one or a combination of the following qualitative indicators of infection and disease in addition to SIBT: ruffled fur, lethargy, loss of appetite, reduced bowel activity, and accumulation of dried nasal mucus. The time from exposure to SIBT and exposure to death was a function of inhaled dose for both Bcbva CA and CI (P <0.0001; S1 Fig); however, dose did not influence time between SIBT and death for either isolate (S2 Fig). For comparative purposes, animals that survived the entire observation period were represented by 500 h data points. Bcbva CA and CI displayed a significantly larger range for time to death, and at lower doses, the median time to death also occurred over a longer range of time, compared to those times reported for *B. anthracis* Ames [25,30] (Table 1).

All animals that succumbed to aerosol exposure were positive for culturable Bcbva CA or CI in blood, lung, liver, spleen, and BAL at death/humane euthanasia (Fig 5 and Table 2), while no culturable bacteria were obtained from animals that survived to the end of the 28-day observation period. Bcbva CA and CI bacterial burdens in terminal tissue and blood samples were at least $10^7$ CFU/g or mL, respectively. Bacteremia could be observed as early as two days prior to death, but in general, bacteremia was not observed until one day prior to death (Fig 5).

## Characterization of Bcbva *in vivo* virulence factor expression

Mean times to SIBT or death (S1 Fig), time to onset of SIBT (S2 Fig) and Bcbva colony phenotypes (S3 Fig) were catalogued during inhalation studies. During the quantification of aerosol samples and, subsequently, bacterial burden, it was observed that both Bcbva isolates displayed differing colony morphologies (S3 Fig). Prior to and following aerosolization, colonies from cultured spores displayed their typical crushed glass and mucoid morphology. When bacteremia was detected at relatively low levels (<~5000 CFU/mL) 1–2 days prior to animal death, large matte colonies reflective of swarming motility were observed (all of the colonies from these samples displayed this morphology). By contrast, colonies derived from terminal blood and tissues (collected from animals that succumbed or euthanized due to infection) were generally highly mucoid of varying size, but round matte colonies could be observed at a <1% frequency. All colony morphologies were PCR positive for *pagA* (pXO1 marker), *capA* (pXO2 marker), and the Genomic Island IV (Bcbva marker), indicating the presence Bcbva carrying both virulence plasmids. When atypical colonies were subsequently passaged, all colonies reverted to the typical colony morphology within one passage on SBA. One animal that succumbed to Bcbva CA had a terminal blood sample that also contained a non-mucoid and *capA* negative colony. Though rare, all other non-mucoid colonies derived from terminal blood samples were *capA* positive. When the stock Bcbva CA frozen stock was screened for *capA*, < 1% of the population was negative, however, all colonies produced a capsule because of the presence of a second hyaluronic acid capsule [16].

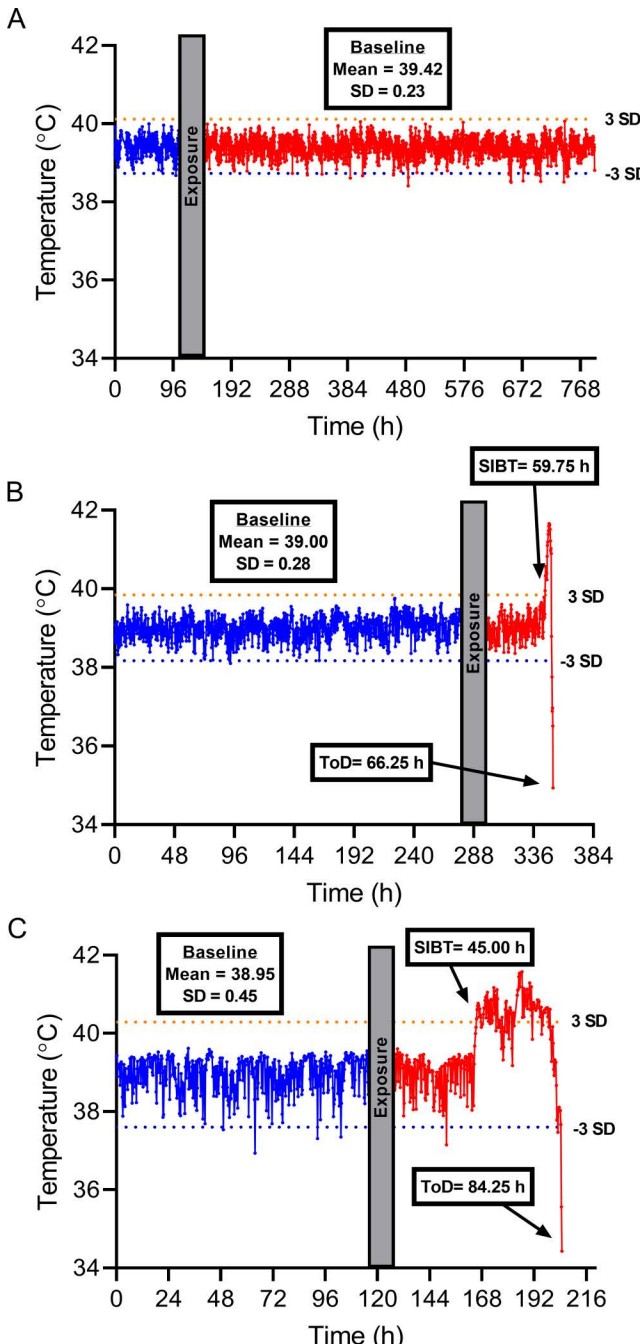

**Fig 3. Representative temperature responses in rabbits exposed by inhalation to Bcbva CI spores.** Significant increase in body temperature (SIBT) and Time of Death (ToD) are indicated with arrows and the value is provided. SIBT is defined as 3x the standard deviation of the baseline temperature. The orange and blue dotted lines indicate 3x the standard deviation above and below the baseline temperature, respectively. Blue temperature readings are the baseline measurements prior to exposure, and red temperature measurements are following exposure. SD indicates standard deviation. **A.** Survivor, dose 2.79 X $10^5$ CFU. **B.** Non-survivor with a short duration between SIBT and death, dose 3.63 X $10^6$ CFU. **C.** Non-survivor with a long duration between SIBT and death, dose 4.31 X $10^5$ CFU.

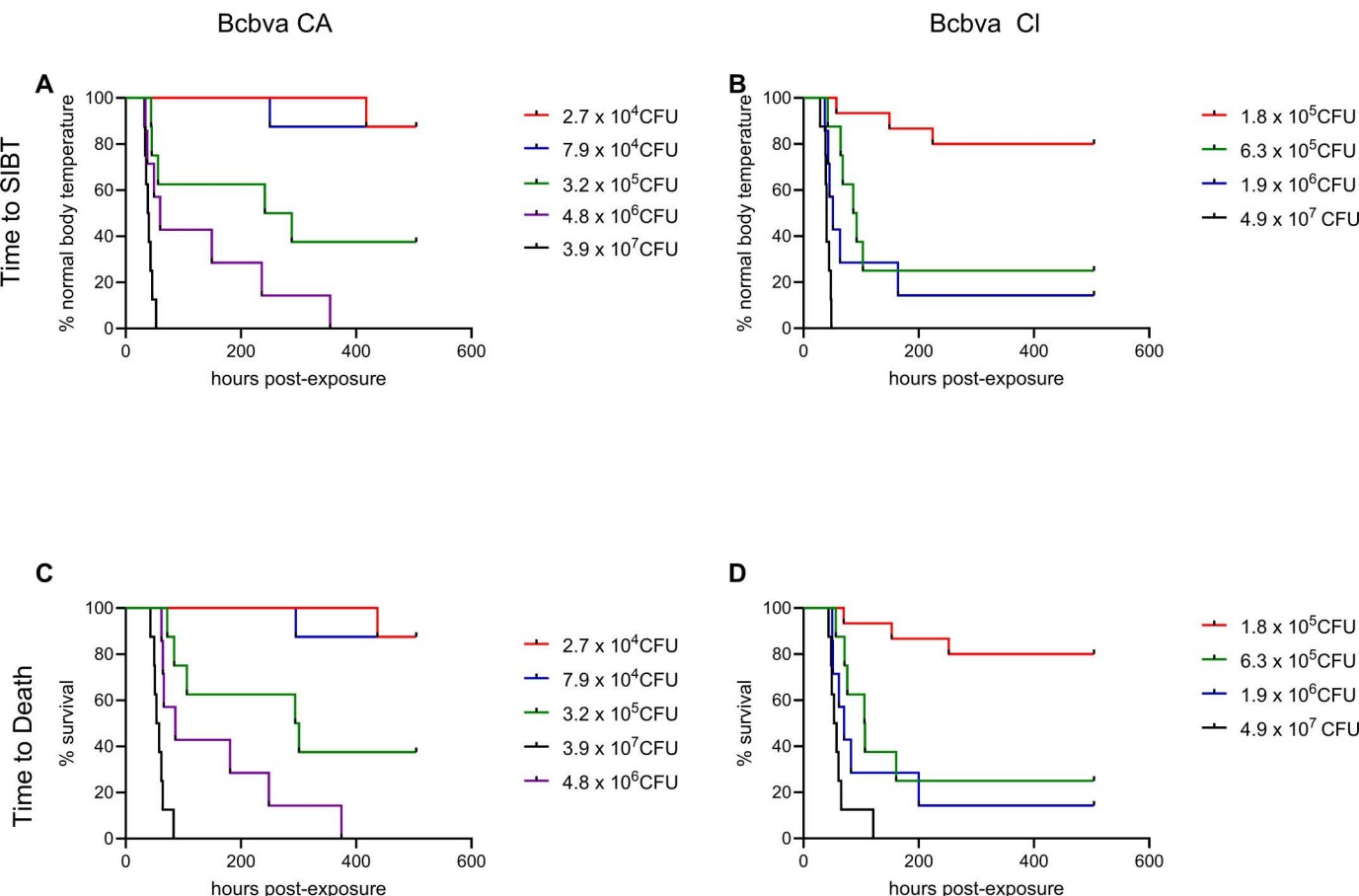

**Fig 4. SIBT and time to death in rabbits exposed by inhalation to Bcbva CA and CI spores.** The time to SIBT following aerosol exposure was recorded for each animal and plotted as the percentage of animals with normal body temperature per dosage group (**A**, **B**). The time of death for animals that succumbed to infection was plotted as percent survival per dosage group (**C**, **D**). Two groups of 8 animals were challenged with 1.8 X 10⁵ CFU Bcbva CI. These groups were combined for data analysis (16 animals in this group).

**Table 1. Dose-dependent median time to death following inhalation exposure to Bcbva CA and CI.** Log-Rank Mantel Cox Test was performed to identify differences in time to death among the administered doses. Data were derived from 8 animals per dose of Bcbva CA and CI and 10 animals for Ames A2084. Since multiple deaths/dose group are required to determine a median time to death, only the data for those doses are listed. The *B. anthracis* data were derived from a previous study [25]. Since dose does not influence time to death for *B. anthracis* challenge, the dose range and mean time to death are reported.

| Bcbva CA | | Bcbva CI | | *B. anthracis Ames* A2084 [25] | |
|---|---|---|---|---|---|
| Dose (CFU) | Median time to Death (h) | Dose (CFU) | Median time to Death (h) | Dose (CFU) | Median time to Death (h) |
| $3.9 \times 10^7$ | 56 | $4.9 \times 10^7$ | 40 | $1.5 \times 10^5 - 1.2 \times 10^6$ | 54 |
| $4.8 \times 10^6$ | 86 | $1.9 \times 10^6$ | 51 | | |
| $3.2 \times 10^5$ | 297 | $6.3 \times 10^5$ | 89 | | |

PA, and HA and PDGA capsule were detected in serum from animals that succumbed to Bcbva CA or CI and the terminal serum PA concentration was consistent with that previously reported for *B. anthracis* in the NZW rabbit model [31]. Evaluation of serial serum samples from both survivors and non-survivors indicated that HA and PA could not be reliably detected prior to animal death, while neither antigen could be detected in survivors during the first five days of the observation period after aerosol exposure (Figs 6 and 7). There was

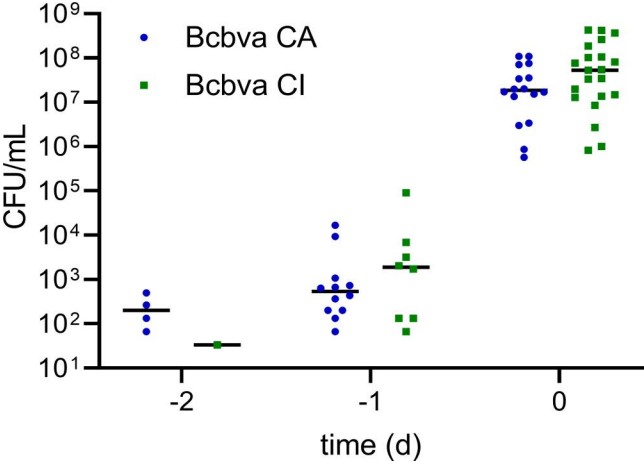

**Fig 5. Longitudinal evaluation of bacteremia following Bcbva inhalation spore challenge.** Symbols indicate each data point, and time points are relative to the time of death in days (d). Mean titers (horizontal line) are shown. The sampling results are shown for animals with at least two consecutive days of sampling (n = 16 for CA; **n** = 21 for CI). Only quantifiable titers are shown. The absence of a data point indicates no detectable bacterial growth observed from the sample of corresponding animals.

**Table 2. Terminal bacterial burden in animals infected with Bcbva.** Animals that succumbed to infection in the LD$_{50}$ studies (n = 23 for CA; n = 25 for CI) were sampled, and mean CFU/ml or g determined for blood, BAL, liver, lung, and spleen tissues. Six survivors were sampled as controls and bacteria were not observed in any of the assayed tissues or fluids at 28 days post-exposure. Data shown are mean with lower and upper 95% confidence intervals in brackets.

| Tissue | Bcbva CA | Bcbva CI | Units |
|---|---|---|---|
| Blood | $3.2 \times 10^7$ [$1.8 \times 10^7$, $4.6 \times 10^7$] | $1.0 \times 10^8$ [$5.2 \times 10^7$, $1.6 \times 10^8$] | CFU/mL |
| BAL | $2.3 \times 10^5$ [$7.1 \times 10^3$, $4.5 \times 10^5$] | $2.2 \times 10^5$ [$8.7 \times 10^4$, $3.5 \times 10^5$] | |
| Liver | $1.2 \times 10^8$ [$6.6 \times 10^7$, $1.7 \times 10^8$] | $1.8 \times 10^8$ [$1.0 \times 10^8$, $2.5 \times 10^8$] | CFU/g |
| Lung | $1.3 \times 10^8$ [$5.2 \times 10^7$, $2.1 \times 10^8$] | $2.6 \times 10^8$ [$1.2 \times 10^8$, $4.0 \times 10^8$] | |
| Spleen | $7.9 \times 10^7$ [$4.1 \times 10^7$, $1.2 \times 10^8$] | $1.9 \times 10^8$ [$9.7 \times 10^7$, $2.8 \times 10^8$] | |

no statistical difference in serum PA, HA, and PDGA concentrations between animals that succumbed in <5 days as compared to those that succumbed later in the observation period. However, in all cases except for Bcbva CA PA expression, in vivo expression of these virulence factors was significantly higher than in vitro culture expression (Fig 8) and was most evident for HA expression, which was > 2 Logs higher than in vitro conditions.

## Assessment of MCM efficacy and disease presentation/progression

NZW rabbits were challenged with a target dose of approximately 200 INHLD$_{50}$s (actual doses: 165.0 and 197.9 INHLD$_{50}$s for Bcbva CA and CI, respectively) and administered pre-exposure or post-exposure MCM as described in Materials and Methods. MCM efficacy was evaluated over a 28-day observation period. As expected, all untreated control animals succumbed to Bcbva CA or CI infection (Table 3 and Figs 9 and 10). All treatment regimens were effective against aerosol exposure to either Bcbva CA or CI. However, after cessation of PEP levofloxacin treatment in animals without vaccination, most animals succumbed to disease with only three animals (1 Bcbva CA and 2 Bcbva CI) surviving through the post-treatment observation period. All animals that received AVA treatment either with or without PEP survived the duration of the post-treatment observation period.

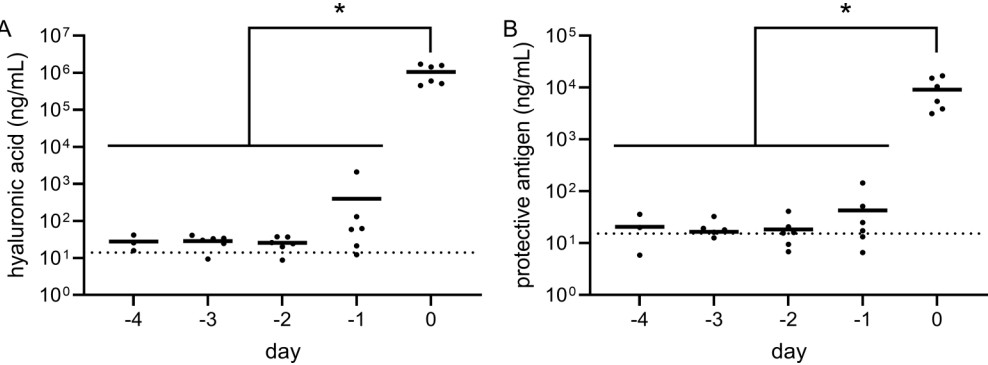

**Fig 6.  Longitudinal evaluation of Bcbva CA virulence factor expression.** Hyaluronic acid (**A**) and protective antigen (**B**) concentrations were measured in serum samples each day for five days following exposure to Bcbva CA spores. Symbols indicate individual data points. Horizontal line indicates the mean. Time points are relative to the time of death (d) for animals that succumbed to aerosol exposure within the first five days of the observation period. The dashed horizontal lines indicate the mean virulence factor expression levels in all samples from survivors for the first 5 days following aerosol exposure, since no statistical increase was observed for survivors during this period. For non-survivors, statistical significance of the level of virulence factor expression was determined using ANOVA analysis followed by a Tukey's multiple comparisons test as described in the methods. * indicates a P-value of< 0.05.

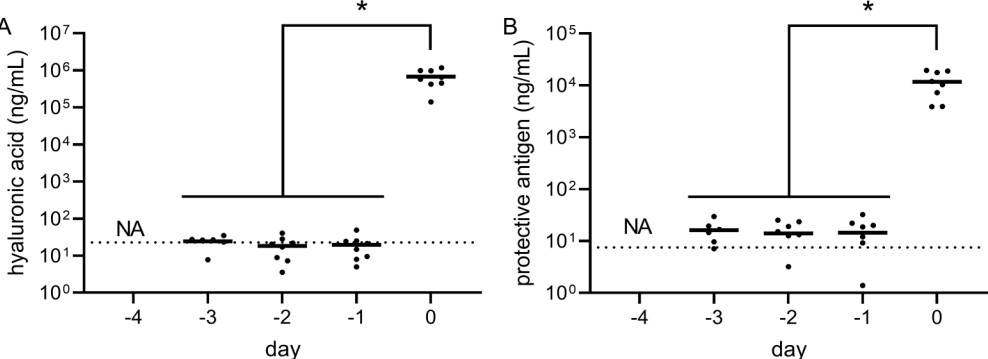

**Fig 7.  Longitudinal evaluation of Bcbva CI virulence factor expression.** Hyaluronic acid (**A**) and protective antigen (**B**) concentrations were measured in serum samples each day for five days following exposure to Bcbva CI spores. Symbol indicates data point. Horizontal line indicates the mean. Time points are relative to the time of death (d) for the analyzed animals that succumbed to aerosol exposure within the first five days of the observation period. The dashed horizontal lines indicate the mean virulence factor expression levels in all samples from survivors for the first 5 days following exposure since no statistical increase was observed for survivors during this period. For non-survivors, statistical significance of the level of virulence factor expression was determined using ANOVA analysis followed by a Tukey's multiple comparisons test as described in the methods. * indicates a P-value of < 0.05.

The primary host phenotypic marker for Bcbva infection is SIBT. As expected, all non-treated animals displayed SIBT prior to succumbing to Bcbva CA or CI infection (Table 3 and Figs 9 and 10). All animals that were pre-vaccinated with AVA (did not receive PEP), or that received only PEP levofloxacin without AVA, exhibited SIBT and 72.2% of animals receiving the dual treatment (AVA and PEP) exhibited SIBT. Further analyses of host temperature responses indicated that there were statistically significant differences in SIBT duration between the PEP levofloxacin and Anthrax Vaccine Adsorbed (pre-AVA) treatment groups, the number of SIBTs between pre-AVA treatment groups, and time to death for all treatments following inhalation spore challenge with either Bcbva CA or CI (Figs 9 and 10, respectively).

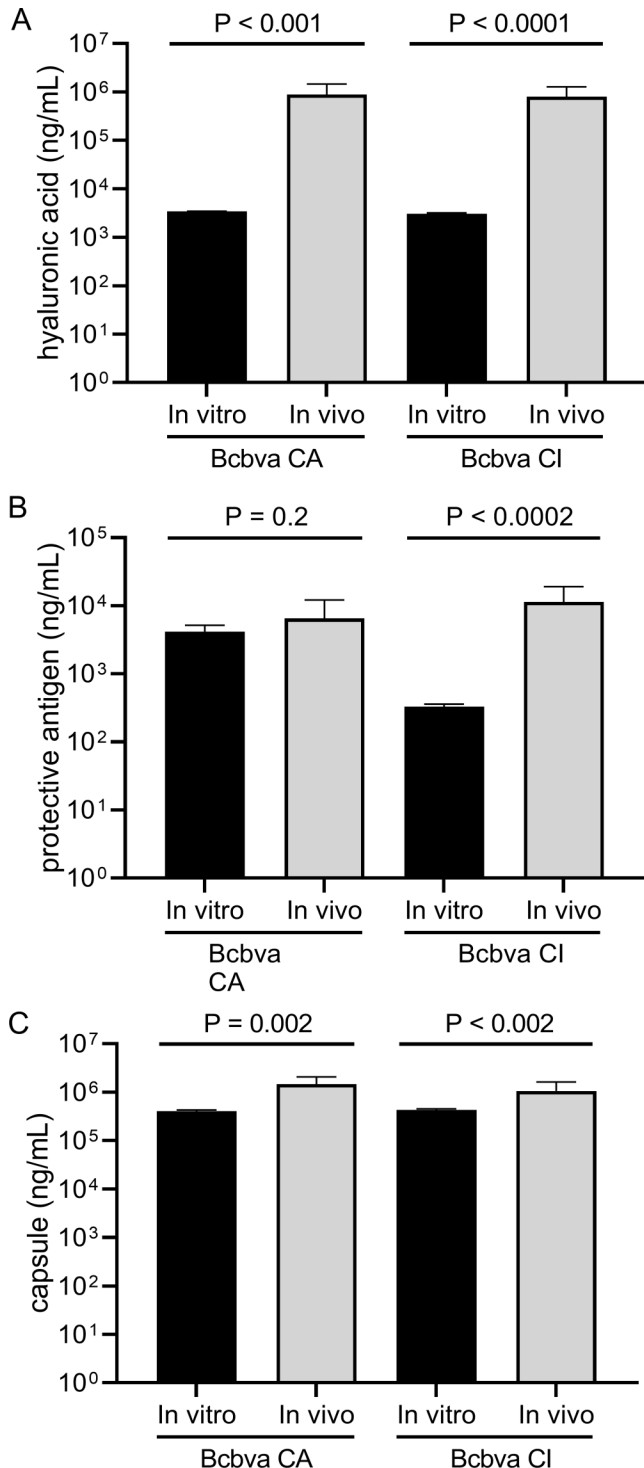

**Fig 8. *In vitro* versus terminal *in vivo* virulence factor expression.** Data represents the mean and standard deviation of the analyzed biological replicates (three replicates for *in vitro*; 10 Bcbva CA and 14 Bcbva CI replicates each for *in vivo*) for hyaluronic acid (A), protective antigen (B), and PDGA capsule (**C**) at stationary growth (*in vitro*) or in terminal blood samples (*in vivo*), which would constitute maximal growth in the respective conditions. The amount of HA in the *B. anthracis* samples was not statistically different from a negative control. HA was not detected in serum collected from animals prior to challenge with Bcbva CA or CI.

**Table 3. Summary of MCM efficacy and outcomes following Bcbva CA or CI aerosol exposure.** MCM were administered as described in Materials and Methods. *Indicates animal succumbed immediately after treatment. Animal data was censored. #Indicates a telemetry transponder failed, and no temperature data was collected. βIndicates SIBT occurred following completion of treatment course. ^Indicates quantified values were not statistically different than pre-administration samples via pair-wise comparison. NA indicates not applicable. †Indicates seroconversion occurred before exposure.

| Strain | Treatment | SIBT | Survival through treatment | Survival 28-day observation period | PA/ HA in non-survivors | PA/ HA in survivors (day 28) | Anti-PA seroconversion of survivors |
|---|---|---|---|---|---|---|---|
| Bcbva CA | Control | 20/20 | NA | 0/20 | Yes/ Yes | No Survivors | NA |
| | PEP Levofloxacin | 20/20β | 20/20 | 1/20 | Yes/ Yes | No^/ No^ | Yes |
| | PEP Levofloxacin & AVA | 13/18*#β | 19/19* | 19/19* | NA | No^/ No^ | Yes |
| | Pre–AVA | 20/20 | 20/20 | 20/20 | NA | No^/ No^ | NA† |
| Bcbva CI | Control | 20/20 | NA | 0/20 | Yes/ Yes | No Survivors | NA |
| | PEP Levofloxacin | 20/20β | 20/20 | 2/20 | Yes/ Yes | No^/ No^ | Yes |
| | PEP Levofloxacin & AVA | 13/18*β | 18/18* | 18/18* | NA | No^/ No^ | Yes |
| | Pre–AVA | 20/20 | 20/20 | 20/20 | NA | No^/ No^ | NA† |

Significantly higher peak temperatures relative to baseline for PEP levofloxacin and pre-AVA treatment groups and increased time to SIBT for all treatment groups were observed following Bcbva CA challenge (Fig 9), while a statistically significant increase in time to SIBT for both PEP treatments was observed following Bcbva CI administration (Fig 10).

Neither Bcbva CA nor CI were detected in serial blood samples through day 7 post challenge in any of the treatment groups. This contrasts with the observation of SIBT in some of the treatment groups above, and as reported previously that *B. anthracis* bacteremia correlates with SIBT in this animal model [31]. Bacteria were not detected in serial blood samples from survivors in any treatment group throughout the observation period, or in spleen and liver samples collected on day 28. Bacteria were detected in lung and BAL samples in some surviving animals, but the detection of bacteria within the lung did not correlate with a specific treatment, isolate, or rabbit sex. For non-treated control animals and animals that succumbed following the cessation of levofloxacin treatment, bacteria were quantifiable in all terminal blood and tissue samples (Fig 11). The PEP levofloxacin treatment significantly reduced bacterial burden in terminal blood, lung, liver, and spleen samples from animals that succumbed to Bcbva CI, while bacterial burdens were only reduced in terminal blood and lung samples for animals that succumbed to Bcbva CA.

## Virulence factor detection and response with MCM

Both HA and PA could be detected in terminal samples for untreated animals and animals that succumbed to Bcbva CA or CI following the cessation of levofloxacin-only treatment (Fig 12). Post-exposure levofloxacin treatment alone resulted significantly lower HA and PA in serum from animals exposed to Bcbva CA but not in animals exposed to Bcbva CI. Neither HA nor PA could be quantified above background in animals treated with AVA vaccination with or without PEP levofloxacin and exposed to either strain.

Evaluation of host immune response was performed measuring seroconversion, defined here as a 3-fold increase in serum anti-PA83 IgG above the background mean. For PEP levofloxacin-treated animals, all survivors (three between Bcbva CA and CI exposed animals) seroconverted by day 28 (Fig 13A), and 28.6% of animals that succumbed following the cessation of PEP levofloxacin treatment seroconverted, but with an IgG titer that was 2-Logs lower than survivors. All evaluated PEP levofloxacin and AVA treated animals seroconverted by day 28, and no statistical difference in IgG titer was observed among survivors as a function of MCM treatment. PEP antibiotic only treated survivors (CA and CI exposures grouped together = three) had antibody levels equivalent to those observed following vaccination with AVA.

All untreated control animals in MCM studies died before seroconversion would have occurred and were not included in seroconversion analysis. Therefore, characterization of

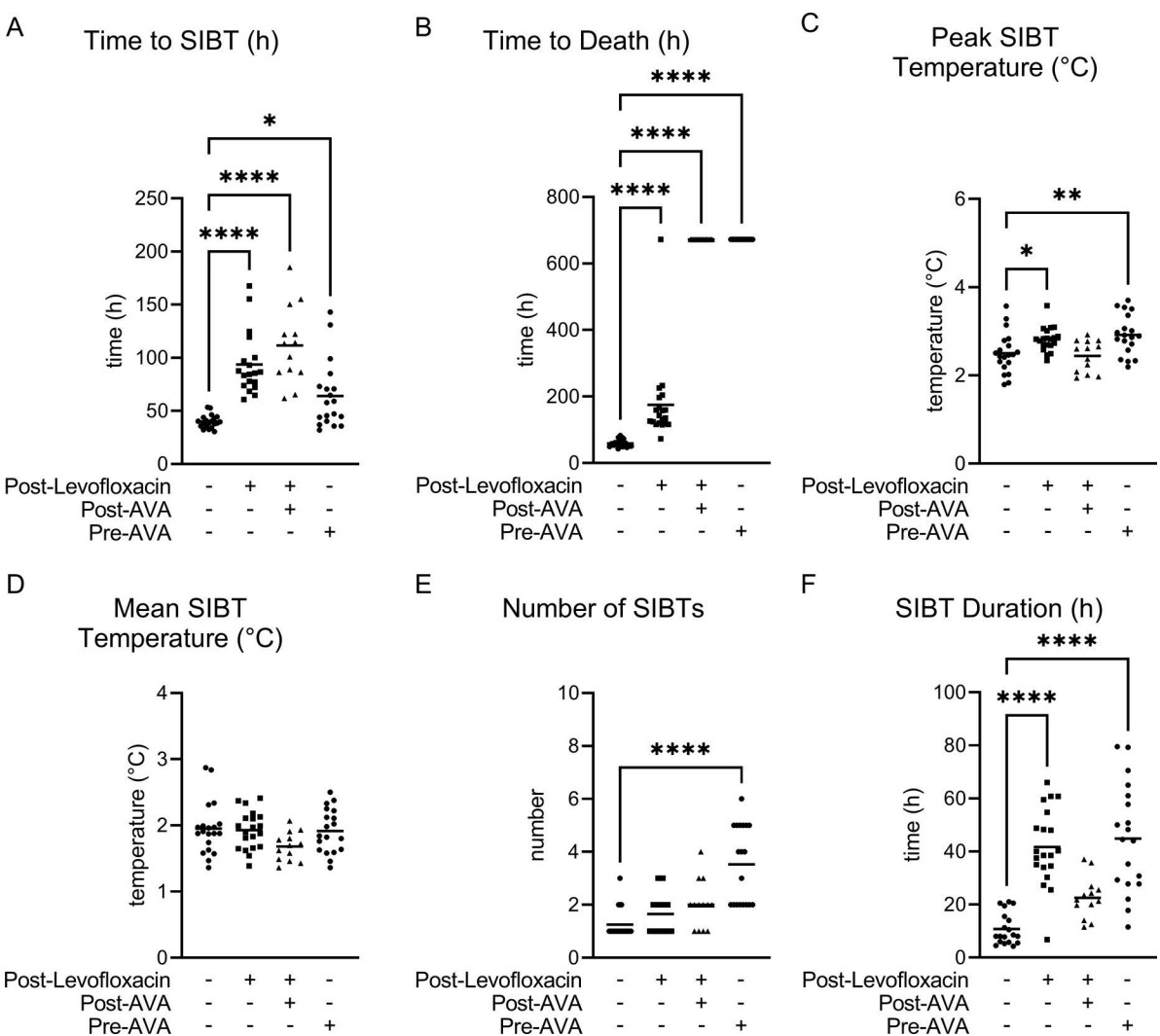

**Fig 9. Temperature responses and time to death as a function of treatment following Bcbva CA aerosol exposure.** Time to SIBT (A), time to death (B), magnitude of SIBT (peak and mean; C and D, respectively), number of SIBT (E), and cumulative duration of SIBT (F)] are indicated. Post indicates post-exposure prophylaxis (PEP). The time to SIBT for post levofloxacin-treated animals was determined from cessation of treatment. If the animal did not display SIBT or survived the 28-day observation period, 672.00 h was used as the time to SIBT and death, respectively. + Indicates treatment occurred. – Indicates treatment did not occur. One-way ANOVA was performed followed by a Dunnett's Test to perform pair-wise comparisons between the MCM treatment groups and control. * Indicates $P < 0.05$. ** Indicates $P < 0.01$. *** Indicates $P < 0.001$. **** Indicates $P < 0.0001$. Only statistically significant comparisons are shown.

seroconversion of naïve animals used samples from INHLD$_{50}$ studies due to the lower exposure doses of Bcbva CA and CI resulting in mixed populations of survivor and non-survivors necessary for these analyses (Fig 13B). Anti-PA IgG levels from both survivors and non-survivors of either Bcbva CA or CI exposed animals were not detectable above background.

## Discussion

This study determined the lethal doses and evaluated disease progression in NZW rabbits challenged by aerosol with Bcbva CA or CI spores. Measurements of disease progression included SIBT, bacterial organ burden and virulence factor expression. The results of these studies demonstrate that Bcbva anthrax-like infections are similar to *B. anthracis* in LD$_{50}$, onset of

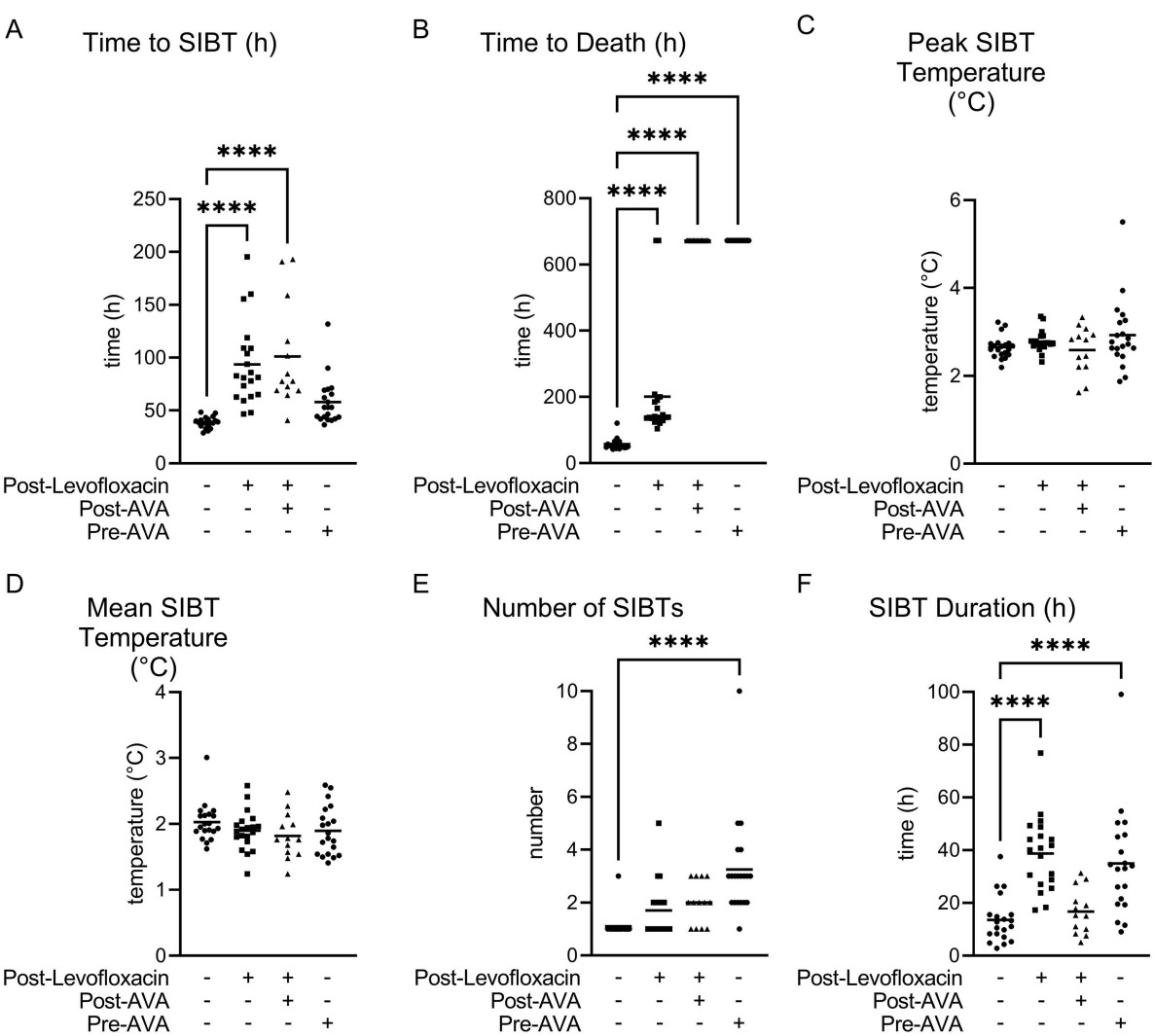

**Fig 10. Temperature responses and time to death as a function of treatment following Bcbva CI aerosol exposure.** Temperature response [time to SIBT (A), time to death (B), magnitude of SIBT (peak and mean; C and D, respectively), number of SIBT (E), and cumulative duration of SIBT (F)] are indicated. Post indicates post-exposure prophylaxis (PEP). The time to SIBT for levofloxacin treated animals was determined from cessation of treatment. If the animal did not display SIBT or survived the 28-day observation period, 672.00 used as the time to SIBT and death, respectively. + Indicates treatment occurred. – indicates treatment did not occur. One-way ANOVA was performed followed by a Dunnett's Test to perform pair-wise comparisons between the MCM treatment groups and control. * Indicates $P < 0.05$. ** Indicates $P < 0.01$. *** Indicates $P < 0.001$. **** Indicates $P < 0.0001$. Only statistically significant comparisons are shown.

SIBT, time to death, bacterial burden, and virulence factor expression. Furthermore, the results indicate that CDC-recommended treatments were effective against aerosol exposure to either Bcbva CA or CI. No treatment fully prevented the appearance of signs of infection, and most infections treated with PEP antibiotic alone resulted in death after cessation of treatment.

When comparing results of Bcbva CA and CI inhalation exposure reported here with previous *B. anthracis* infection studies that utilized the NZW rabbit model, there was no statistical difference in median infectious dose (as determined by SIBT) or lethal dose [25,29]. Consistent with the *B. anthracis* NZW rabbit model, all animals that succumbed following aerosol exposure to Bcbva previously exhibited SIBT, and all animals that exhibited SIBT succumbed to Bcbva [25]. Additionally, no sex bias was observed for SIBT or infectious/lethal

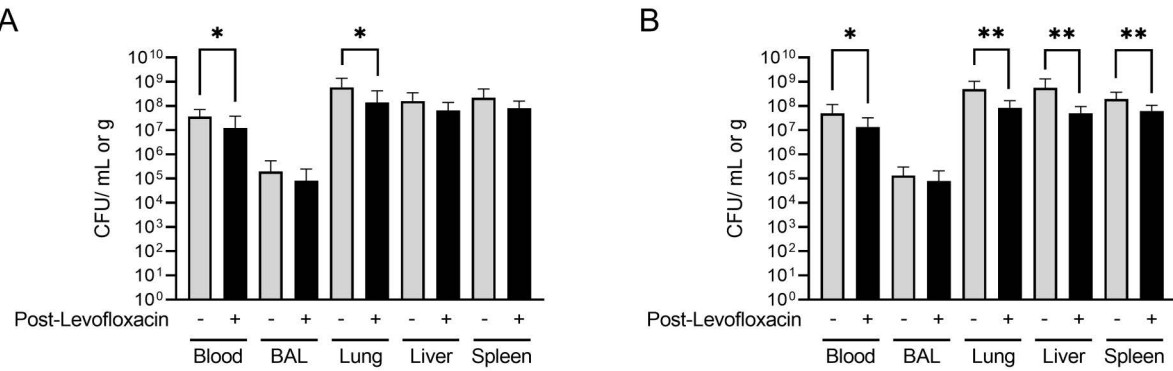

**Fig 11. Terminal bacterial burden in animals that succumbed following the cessation of levofloxacin treatment and non-treated controls.** The CFU/mL in blood or broncho-alveolar lavage (BAL) samples and CFU/g of lung, liver, or spleen tissues of terminal untreated animals and animals that died after completion of levofloxacin treatment were quantified for Bcbva CA (A), or Bcbva CI (B). + Indicates treatment. – Indicates no treatment. Pair-wise comparisons were performed with a 2-way Student's T-test comparing bacterial titers between tissues harvested from levofloxacin-treated animals and control. * Indicates $P < 0.05$. ** Indicates $P < 0.01$. Only statistically significant comparisons are shown.

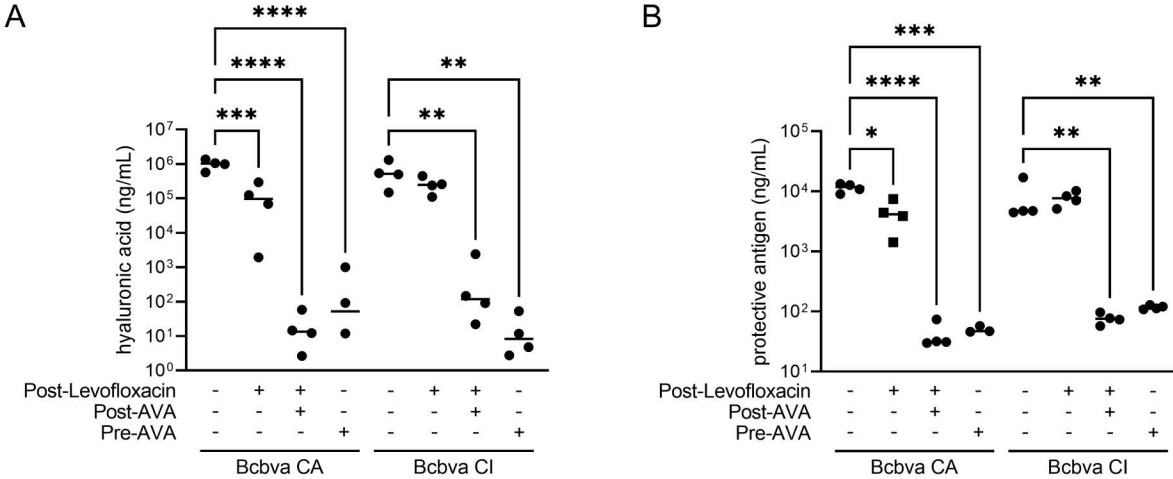

**Fig 12. Quantification of virulence factors in MCM-treated animals following aerosol exposure to Bcbva CA and CI.** Hyaluronic acid (A) and protective antigen (B) were quantified in serum samples upon death or from survivors at the end of the observation period. Symbols indicate individual data points. Horizontal line indicates the mean. For control and levofloxacin-only treated animals the data values are from non-survivor terminal samples. For dual treatment with PEP and Post-AVA, and pre-AVA animals, the data values represent quantification of day 28 samples since no animals succumbed during the observation period. Collectively, comparisons were made between survivors and non-survivors as a function of treatment regimen. + Indicates treatment. – Indicates no treatment. One-way ANOVA was performed followed by a Tukey's Test to perform pair-wise comparisons between the MCM treatment groups and control. * Indicates $P < 0.05$. ** Indicates $P < 0.01$. *** Indicates $P < 0.001$. **** Indicates $P < 0.0001$. Only statistically significant comparisons are shown.

dose. Like *B. anthracis*, SIBT preceded or coincided with other clinical signs of infection, specifically the detection of PA in the sera and bacteria in the blood [31, 32]. Furthermore, the presence of PA and HA in the sera only occurred when quantifiable bacteria were present within blood samples. Lastly, as with *B. anthracis* infections, terminal tissue sampling demonstrated systemic Bcbva dissemination within the rabbit with $10^6$ - $10^8$ CFU/mL of blood or gram of tissues (lung, liver, and spleen) [29,33,34], with few outward signs of infection or disease observed prior to death.

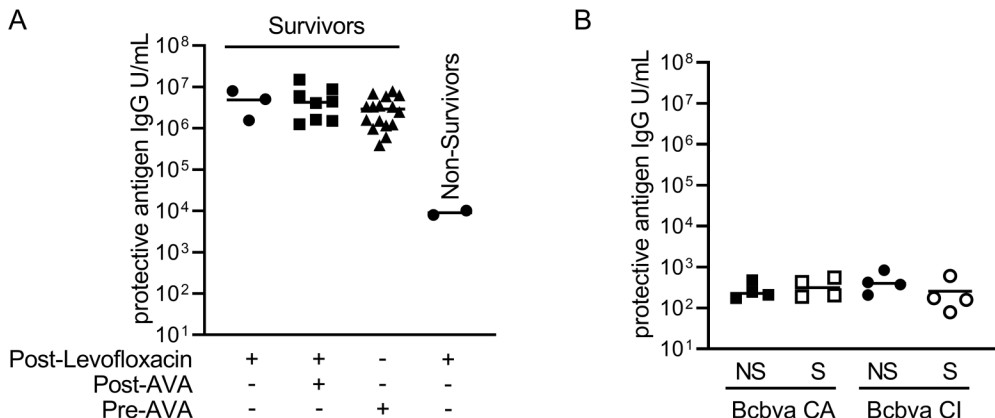

**Fig 13. Seroconversion against protective antigen.** Antibodies that bind to protective antigen were measured in serum samples from rabbits exposed to Bcbva and treated with medical countermeasures (A). Seroconversion in MCM-treated animals. Only the animals that seroconverted are shown. Results for Bcbva CA and CI experiments were combined since a statistical difference was not observed as a function of challenge strain by ROUT outlier analysis (post-levofloxacin survivors) and pair-wise comparisons with a two-way Student's T-test (post-levofloxacin and AVA; Pre-AVA). The survival outcome for each treatment group is indicated. + Indicates treatment occurred. – Indicates treatment did not occur. Symbols indicate individual data points. Horizontal line indicates the mean. One-way ANOVA was performed followed by a Tukey's Test to perform pair-wise comparisons between the MCM treatment groups, except for the post-levofloxacin non-survivor group (too few data point for statistical analysis). Only statistically significant comparisons are shown. Blood serum samples from $INHLD_{50}$ experiments were quantified for antibody that binds protective antigen (B). Pair-wise comparisons were performed with a 2-way Student's T-test comparing survivors and non-survivors. NS indicates non-survivors. S indicates survivors.

There are several notable differences in disease progression when comparing Bcbva to *B. anthracis*. For *B. anthracis*, NZW rabbits usually succumb between 2–5 days following inhalation dosing with rare exception [25,30], and there is not a dose-dependent time to SIBT or death. Most animals that succumbed to Bcbva challenge succumbed between 2–5 days, but several animals succumbed outside of this range for both Bcbva CA and CI, including one animal succumbing after 18 days. Furthermore, time to SIBT and death were observed to be dose-dependent for both Bcbva isolates, with higher doses resulting a shorter median time to event (S1 Fig). This has been reported for *B. anthracis* [35] but is not consistently observed. It is possible that motility of Bcbva strains may contribute to this observation of dose dependence but this would require further investigation. We observed some degree of motility for Bcbva by wet mount microscopy.

In previous studies, colony morphology of *B. anthracis* derived from blood or tissue samples remains unchanged regardless of the sample source, duration of the infection, infection model, or standard *in vitro* culturing (in the absence of carbon dioxide and bicarbonate in culture media) [25]. By contrast, Bcbva displayed a polymorphic colony morphology that depended on the time after challenge that the sample was taken, and the type of tissue sampled. As noted above, Bcbva can display motility in contrast to *B. anthracis* which is non-motile due to mutations in the flagellar gene cluster and as such would not be able to display morphological variation based on motility. The observation that these Bcbva morphological differences are lost after a single passage suggests that they are a function of differential gene regulation *in vivo*. Quorum sensing or an alternative bacterial density signaling process may play a role in the transition from a swarming motility colony morphology to a mucoid colony, because the swarming motility colony morphology was only observed when bacteremia was in relatively low numbers in blood samples. This dynamic morphological distribution could potentially confound diagnostic evaluation of Bcbva infections, as the swarming motility colony morphology

and highly mucoid colony morphologies are quite distinct from the shiny crushed glass colony morphology that is typical of Bcbva cultures grown in vitro. This further supports the necessity for molecular or genomic diagnostic platforms for positive identification of Bcbva infections if pX02 and PA seroconversion ELISA reagents are not available [20,36].

A previous study determined the lethality and disease progression of Bcbva CA and CI using subcutaneous and intranasal mouse infection models, and a subcutaneous guinea pig model [17]. The median lethal doses for both Bcbva isolates were approximately 1 Log higher in a NZW rabbit inhalation model as compared to mouse intranasal studies, and this difference is consistent with similar lethality assessments performed with *B. anthracis* using these two models [17,23,25,29,37–39]. Like the intranasal mouse model, Bcbva infection in NZW rabbits resulted in vegetative cell dissemination, with growth observed in distal organs, and similar time to death as the mouse and guinea pig models [17]. It is possible that the presence of PDGA capsule may not be essential during an inhalation anthrax-like infection in NZW rabbits as demonstrated for Bcbva CA intranasally in mice [17]. However, this requires further investigation. In contrast to the previous intranasal mouse study, we did not observe evidence of neurological involvement in NZW rabbits.

The data from the MCM studies demonstrate that pre-exposure vaccination with AVA and CDC-recommended PEP regimens with levofloxacin, with or without AVA, are protective against aerosol exposure to lethal doses of either Bcbva CA or CI. These results are consistent with those observed for *B. anthracis* [40–43]. Prior studies with Bcbva in mice demonstrated that an rPA-only vaccine was not fully efficacious and required the addition of formaldehyde fixed spores to achieve full protection, raising the concern that some PA-based anthrax vaccines may not be as effective against Bcbva. However, the prior study did not evaluate AVA efficacy against Bcbva. The results from the present study indicate that AVA is fully effective against Bcbva challenge as a part of either a pre-or post-exposure treatment regimen. As observed in *B. anthracis* studies with high challenge doses, the cessation of levofloxacin-only treatment results in most animals succumbing to anthrax. Additionally, the treatments did not universally prevent infection and symptom onset, and pre-AVA did not result in sterilizing immunity. Collectively, these results suggest that CDC recommendations for the treatment of inhalation anthrax would be effective against anthrax-like disease caused by Bcbva.

## Supporting information

**S1 Data. Raw data supporting information.**
(PDF)

**S1 Fig. Mean time to SIBT or death following aerosol exposure to Bcbva CA (left) and CI (right).** The time to SIBT (A, B) and death (C, D) are indicated with a symbol. One-way ANOVA was performed followed by a Tukey's multiple comparison test to perform pair-wise comparisons between each does group. Animals that survived the entire observation period were represented by data points at 500 hours post-exposure for comparative purposes. * indicates a P-value of < 0.05. ** indicates a P-value of < 0.01. *** indicates a P-value of < 0.001. **** indicates a P-value of < 0.0001. Two groups of 8 animals were challenged with 1.8 X 10$^5$ CFU Bcbva CI. These groups were combined for data analysis (16 animals in this group).
(TIF)

**S2 Fig. Time between onset of SIBT and death/humane euthanasia.** The individual data points with the mean are plotted for the time between SIBT and death for rabbits exposed to Bcbva isolates CA (A) or CI (B). One-way ANOVA was performed followed by a Tukey's multiple comparison test to perform pair-wise comparisons between each dosage group. No

significant differences were found between groups. Dosage groups 2.7 x 104 and 7.9 x 104 with CA could not be compared due to n = 1.
(TIF)

**S3 Fig. Colony morphologies of Bcbva.** Representative colonies of each colony morphology are shown from pre-aerosol, post-aerosol, and in vivo samples for Bcbva CA and Cl. Heat shocked spores prior to (panel 1) and following (panel 2) aerosolization with a 3-Jet Collison nebulizer. The colonies are average in size, mucoid, and display a fried egg appearance. Pre-terminal bacteremic blood samples (panel 3 and 4). The colonies appear large, matte, round or amorphous, and indicative of swarming motility. The colony in the top right panel (4) was sampled with a loop to demonstrate the tackiness of the colony. Terminal bacteremic blood samples (panels 5–8), which also represent the colony morphologies obtained from lung, liver, and spleen homogenates (have similar appearance). The colonies are average in size and either highly mucoid with fried egg-like appearance (predominant morphology) or matte and round (infrequent). Terminal bacteremic blood samples. The colonies are either average in size and round with matte or highly mucoid presentations or small, round, and mucoid. Images represent colony morphologies observed for both isolates. The images were scaled so that the vertical diameters of agar plates are identical among all images.
(TIF)

## Acknowledgements

We would like to acknowledge the Robert Koch Institute, Berlin, Germany, for providing the Bcbva isolates. We would also like to thank Stephanie Hannah for critical review of the manuscript and thank Donald Chabot and Arthur Friedlander for providing the labeled and un-labeled PDGA antibody and protocols. The views and conclusions contained in this document are those of the authors and should not be interpreted as necessarily representing the official policies, either expressed or implied, of the DHS or the U.S. Government. DHS does not endorse any products or commercial services mentioned in this document.

*Disclaimer*: This manuscript has been authored by Battelle National Biodefense Institute, LLC under Contract No. HSHQDC-15-C-00064 with the U.S. Department of Homeland Security. The United States Government retains and the publisher, by accepting the article for publication, acknowledges that the United States Government retains a non-exclusive, paid up, irrevocable, world-wide license to publish or reproduce the published form of this manuscript, or allow others to do so, for United States Government purposes.

## Author contributions

**Conceptualization:** Victoria Wahl, Angelo Scorpio.

**Data curation:** Allison M. Ferris, David G. Dawson, Andrea B. Eyler, John J. Yeager, Jordan K. Bohannon, Jeremy A. Boydston, Melissa L. Krause, Charles L. Balzli, Tammy D. Jenkins, Sherry L. Rippeon, James E. Miller, Susan E. Miller, David W. Clark, Emmanuel Manan, Ashley F. Harman, Kim R. Rhodes, Tina M. Sweeney, Heather D. Cronin, Heather A. Zimmerman, Alec S. Hail, Angelo Scorpio.

**Formal analysis:** Angelo Scorpio.

**Funding acquisition:** Victoria Wahl.

**Investigation:** Angelo Scorpio.

**Methodology:** Allison M. Ferris, John J. Yeager, Jeremy A. Boydston, Victoria Wahl, Angelo Scorpio.

**Resources:** Ron L. Bowman, Michael P. Winpigler.

**Supervision:** Victoria Wahl, Angelo Scorpio.

**Writing – original draft:** Angelo Scorpio.

**Writing – review & editing:** Angelo Scorpio.

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
