## [Decision Letter · Decision Letter 0]

14 Feb 2025

PNTD-D-25-00035Bacillus cereus biovar anthracis causes inhalational anthrax-like disease in rabbits that is treatable with medical countermeasuresPLOS Neglected Tropical Diseases Dear Dr. Scorpio, Thank you for submitting your manuscript to PLOS Neglected Tropical Diseases. After careful consideration, we feel that it has merit but does not fully meet PLOS Neglected Tropical Diseases's publication criteria as it currently stands. Therefore, we invite you to submit a revised version of the manuscript that addresses the points raised during the review process. Please submit your revised manuscript within 30 days Apr 15 2025 11:59PM. If you will need more time than this to complete your revisions, please reply to this message or contact the journal office at plosntds@plos.org. Please include the following items when submitting your revised manuscript: * A rebuttal letter that responds to each point raised by the editor and reviewer(s). You should upload this letter as a separate file labeled 'Response to Reviewers '. This file does not need to include responses to any formatting updates and technical items listed in the 'Journal Requirements' section below. * A marked-up copy of your manuscript that highlights changes made to the original version. You should upload this as a separate file labeled 'Revised Manuscript with Track Changes '. * An unmarked version of your revised paper without tracked changes. You should upload this as a separate file labeled 'Manuscript '. If you would like to make changes to your financial disclosure, competing interests statement, or data availability statement, please make these updates within the submission form at the time of resubmission. Guidelines for resubmitting your figure files are available below the reviewer comments at the end of this letter. We look forward to receiving your revised manuscript. Kind regards, Adly M.M. Abd-Alla, Prof asso.Academic EditorPLOS Neglected Tropical Diseases Stuart BlacksellSection EditorPLOS Neglected Tropical Diseases

Shaden Kamhawi

co-Editor-in-Chief

Paul Brindley

co-Editor-in-Chief

**Journal Requirements:**

At this stage, the following Authors/Authors require contributions: Allison M. Ferris, David G. Dawson, Andrea B. Eyler, John J. Yeager, Jordan K. Bohannon, Jeremy A. Boydston, Melissa L. Krause, Charles L. Balzli, Victoria Wahl, Tammy D. Jenkins, Sherry L. Rippeon, James E. Miller, Susan E. Miller, David W. Clark, Emmanuel Manan, Ashley F. Harman, Kim R. Rhodes, Tina M. Sweeney, Heather D. Cronin, Ron L. Bowman, Michael P. Winpigler, Heather A. Zimmerman, Alec S. Hail, and Angelo Scorpio. Please ensure that the full contributions of each author are acknowledged in the "Add/Edit/Remove Authors" section of our submission form.

2) We noticed that you used the phrase 'data not shown' in the manuscript. We do not allow these references, as the PLOS data access policy requires that all data be either published with the manuscript or made available in a publicly accessible database. Please amend the supplementary material to include the referenced data or remove the references.

- ® on Line: 127.

Potential Copyright Issues:

- Please confirm that you are the photographer of Figure S3, or provide written permission from the photographer to publish the photo under our CC BY 4.0 license.

- Figure 1. Please confirm whether you drew the images / clip-art within the figure panels by hand. If you did not draw the images, please provide a link to the source of the images or icons and their license / terms of use; or written permission from the copyright holder to publish the images or icons under our CC BY 4.0 license. Alternatively, you may replace the images with open source alternatives. See these open source resources you may use to replace images / clip-art:

5) Please ensure that the funders and grant numbers match between the Financial Disclosure field and the Funding Information tab in your submission form. Note that the funders must be provided in the same order in both places as well.

**Reviewers' comments:** Reviewer's Responses to Questions

**Key Review Criteria Required for Acceptance?**

**Methods** :

-Are the objectives of the study clearly articulated with a clear testable hypothesis stated?

-Is the study design appropriate to address the stated objectives?

-Is the population clearly described and appropriate for the hypothesis being tested?

-Is the sample size sufficient to ensure adequate power to address the hypothesis being tested?

-Were correct statistical analysis used to support conclusions?

-Are there concerns about ethical or regulatory requirements being met?

Reviewer #1: The study objective and the design were clear enough but quite similar to the work of Ionin et al 2013. The statistical analysis used is correct, the facility where the work was done was accredited and the experiments performed according to standard guidelines. However, the following required response from the authors;

1. The cultivation of the study isolate, B. cereus biovar anthracis as described in line 104 should require biosafety protocols given the virulence status of the bacterium. What were the precautions (IPC) put in place?

2. How were the euthanized rabbits disposed of (line 152)?

3. There is need to sufficiently explain how the incoming air into the 3-jet collison chamber was adequate enough such that low/poor Oxygen threshold would not have contributed to the death of the animals.

4. What anthrax-like symptoms were presented by the animals?

5. Was the ELISA direct or indirect?

Reviewer #2: In general, the methods are well described, with one exception (see comments).

Reviewer #3: Mostly, see specific comments

**Results** :

-Does the analysis presented match the analysis plan?

-Are the results clearly and completely presented?

-Are the figures (Tables, Images) of sufficient quality for clarity?

Reviewer #1: The authors presented the results in tables and images that are of sufficient quality although clearer explanations are needed on the operation of spore delivery instruments.

Reviewer #2: Overall, the results are clearly and completely presented. However, there are some issues which need clarification or can be improved (see comments).

Reviewer #3: Mostly, some figures are overly complex and could use some clarification, others some consolidation.

**Conclusions** :

-Are the conclusions supported by the data presented?

-Are the limitations of analysis clearly described?

-Do the authors discuss how these data can be helpful to advance our understanding of the topic under study?

-Is public health relevance addressed?

Reviewer #1: The authors did not draw any specific conclusions to underline the study findings. The conclusion should be presented as a separate sub-heading. The limitations of analysis should also be distinctly described.

Reviewer #2: The conclusions are supported by the data presented. The main outcome of the study is the fact that there are usually no differences between inhalational infections with Bcbva and B. anthracis, also concerning the efficacy of MCM. However, two differences were observed which should be discussed in more detail (see comments).

Reviewer #3: Yes, a little more regarding previous animal models and other modes of vaccinations should be dicussed.

**Editorial and Data Presentation Modifications?**

Reviewer #1: Minor revision

Reviewer #2: Methods:

The growth conditions to assess the in vitro expression of virulence factors (HA, PA, PDGA) should be added: medium, incubation time, with or without CO2?

Results:

Fig. 2 A and B: The curves are wrong. The highest infection doses result in highest probability of survival, this cannot be true.

Lines 285 and 287: The authors probably refer to Fig. 3 and Fig. S2, not to Fig. 4 and Fig. S1.

Fig. 3: The exposure doses should be indicated in the figure or in the legend. The figure would be easier to understand if the hours would be counted from the timepoint of exposure (like SIBT and ToD) and not from the start of temperature registration.

Fig. 4 D: 5/8 animals (Fig. S1) infected with Bcbva CI survived. This would be 62.5% and not 80% as indicated in the figure.

Table 2: What are the numbers in brackets?

Line 350: it would be better to write “colonies from cultured spores” instead of “cultured spores”.

Line 352: did all colonies (100%) derived from animals 1-2 days prior to death display the large matte phenotype or was this only observed in a fraction of colonies?

Fig. S3: The caption is difficult to understand. It would be easier if the photos were numbered and the explanation in the caption would refer to the numbers.

Fig. 8: The in vitro growth conditions must be described (see Methods). Due to a mutation in the hasA gene, B. anthracis should not be able to express a HA capsule. Did the authors perform the ELISA with a negative control (medium alone) which also shows this background expression? HA is a component of the extracellular matrix of animal tissues. The high in vivo HA concentration in serum might be the result of the animal and not the bacterial HA. An appropriate control (HA ELISA with serum of non-infected animals) must be performed.

Fig. 12: Only 4 data points per group are shown, but the experiment included 20 animals per group (as shown in Fig. 9 and 10). Did the authors analyze HA and PA expression of only 4 animals per group?

Conclusions:

Lines 562 ff: How do the authors explain the dose-dependent time to SIBT or death in Bcbva in contrast to B. anthracis? There seems to be one report for B. anthracis (ref. 35), but it is not consistently observed.

Lines 567 ff: Bcbva colonies derived from animals 1-2 days prior to death displayed a large matte phenotype indicative of swarming motility. In contrast to Bcbva, B. anthracis is non-motile due to mutations in the flagellar gene cluster and would be unable to display this phenotype, regardless of differential gene expression. This should be discussed. It would be interesting to assess the colony morphology of a non-motile deletion mutant of Bcbva.

Line 587: Why do the authors think that there is “some indication that the presence of PDGA capsule may not be essential during ... infection in NZW rabbits”? This was shown in mice with Bcbva lacking the PDGA capsule. Did the authors perform infections with Bcbva lacking the pXO2 capsule plasmid or is this just a conclusion drawn from the fact that a high concentration of HA was detected in rabbit serum?

Acknowledgements, line 606: please indicate “Robert Koch Institute, Berlin, Germany”.

Reviewer #3: See specific comments

**Summary and General Comments** :

Reviewer #1: (No Response)

Reviewer #2: The study adds to the knowledge of Bcbva pathogenesis in an animal model (NZW rabbits) from which data were not yet available. MCM intervention which is used for treatment of inhalational anthrax caused by B. anthracis seems to be effective also against Bcbva.

Reviewer #3: This is a well written paper full of data describing infections and AVA vaccine protection against two emerging anthrax-causing Bcbva strains.

PLOS authors have the option to publish the peer review history of their article (what does this mean? ). If published, this will include your full peer review and any attached files.

**Do you want your identity to be public for this peer review?** For information about this choice, including consent withdrawal, please see our Privacy Policy .

Reviewer #1: **Yes: ** Bamidele Tajudeen Akanji

Reviewer #2: No

Reviewer #3: No

---

## [Editor Report · Decision Letter 1]

11 Mar 2025

Dear Dr. Scorpio,

We are pleased to inform you that your manuscript 'Bacillus cereus biovar anthracis causes inhalational anthrax-like disease in rabbits that is treatable with medical countermeasures' has been provisionally accepted for publication in PLOS Neglected Tropical Diseases.

Best regards,

Adly M.M. Abd-Alla, Prof asso.

Section Editor

Stuart Blacksell

Section Editor

Shaden Kamhawi

co-Editor-in-Chief

Paul Brindley

co-Editor-in-Chief

---

## [Editor Report · Acceptance letter]

Dear Dr. Scorpio,

We are delighted to inform you that your manuscript, "Bacillus cereus biovar anthracis causes inhalational anthrax-like disease in rabbits that is treatable with medical countermeasures," has been formally accepted for publication in PLOS Neglected Tropical Diseases.

Best regards,

Shaden Kamhawi

co-Editor-in-Chief

Paul Brindley

co-Editor-in-Chief
